# Double Shield: The Roles of Personal and Organizational Resources in Promoting Positive Outcomes for Employees During Wartime

**DOI:** 10.3390/ijerph22091384

**Published:** 2025-09-04

**Authors:** Ronit Nadiv, Marianna Delegach

**Affiliations:** Managing Human Resource Department, Sapir Academic College, Ashkelon 7915600, Israel; mariannad@sapir.ac.il

**Keywords:** burnout, COR theory, organizational resources, personal resources, war, wellbeing

## Abstract

Employee well-being is essential for organizational growth and success in stable times and is even more critical during crises and life-threatening events. Although the COVID-19 pandemic highlighted the importance of holistic approaches to sustaining employee well-being, limited research has been conducted to identify strategies for maintaining employee well-being and preventing burnout during life-threatening events, such as wars or terrorist attacks. Addressing this gap, the current study investigates how and why a range of organizational resources (i.e., perceived organizational support, managerial accessibility, and psychological safety) and personal resources (i.e., hope and paradox mindset) contribute to reducing employee burnout in times of existential threat. Drawing on Conservation of Resources (COR) theory, we propose that employee well-being mediates the relationship between organizational and personal resources and burnout at work. Data were collected through an online two-wave survey administered by a professional survey firm with access to a diverse pool of Israeli employees across occupations and work roles in November (time 1) and December 2023 (time 2), following the October 7 terrorist attack by Hamas. A time-lagged design, with key outcomes collected one month after the predictors, was employed to reduce the risk of common method bias. The data were analyzed using path analysis with bootstrapped indirect effects. The results demonstrate that hope, organizational support, psychological safety, and managerial accessibility positively contribute to employee well-being, which, in turn, is associated with lower levels of burnout. Theoretical and practical implications of these results are discussed.

## 1. Introduction

In the 21st century, global crises—both natural and man-made—have increased dramatically, posing significant challenges to individuals, organizations, and societies. Natural disasters such as hurricanes, floods, and earthquakes are becoming more frequent, while human-caused crises, including terrorism and war, remain constant threats to societal stability [1,2]. These events not only disrupt economic and social systems, but also profoundly impact employees’ physical and psychological well-being, leading to cascading consequences for organizational performance [2]. While extensive research has explored employee well-being during natural disasters and the COVID-19 pandemic (e.g., [3,4,5,6]), little is known about how employees navigate crises such as war, which are marked by sustained violence, widespread fear, and prolonged societal instability.

The COVID-19 pandemic exposed critical vulnerabilities in workplaces, underscoring the importance of robust organizational support systems [3,4]. Similarly, studies on crises like 9/11 have revealed that exposure to traumatic events heightens psychological distress, reduces well-being, and increases symptoms of burnout and depression and that organizational and social supports can buffer these effects [7,8]. However, the majority of this research has focused on short-term or localized crises (e.g., terrorist attacks or natural disasters). In contrast, war—marked by prolonged conflict, persistent fear, and widespread societal disruption—presents unique and largely unexplored challenges for organizational resilience and employee well-being [9,10].

Wars differ from other crises in their intentionality, scope, and prolonged impact. Unlike natural disasters, wars and terrorist attacks are deliberate acts aimed at destabilizing communities, threatening both physical safety and mental health [11,12]. The sustained violence and pervasive instability of war result in prolonged economic and social disruption, including work stoppages and job insecurity, which amplify uncertainty for employees [13]. Employees exposed to war face heightened levels of stress, anxiety, and post-traumatic stress disorder [14,15,16]. However, research into mechanisms that could mitigate the depletion of employees’ resources during wartime is still limited [9].

This study aimed to fill this lacuna by focusing on the ongoing military conflict in Israel after the terrorist attack by Hamas on 7 October 2023. The aftermath of this devastating event escalated into a prolonged war, destruction of infrastructure, and heightened levels of trauma among its workforce [15]. While it is important to note that both sides in the conflict have suffered widespread losses of life, we test our hypothesis based on data obtained from the Israeli side of the border due to a lack of access to respondents on the Palestinian side. By situating this study within such a high-stakes environment, we sought to understand how employee well-being and burnout are affected during wartime and how personal and organizational resources can be utilized to mitigate those effects.

Specifically, the study’s contributions are threefold. First, it examines employees’ well-being and burnout during wartime—an understudied external workplace stressor (e.g., [9,13]). This study seeks to fill this gap by drawing on COR theory [17], positing that a reservoir of organizational and personal resources enables employees to maintain their well-being during the war, thereby reducing levels of burnout. Second, we propose an integrated model of personal and organizational resources that can mitigate the negative effects of war on the well-being and burnout of employees. Finally, our study advances the debate on the relationship between well-being and burnout by treating them as interrelated but distinct constructs. Specifically, we examine whether lower levels of well-being are associated with higher levels of burnout, consistent with the idea that diminished resources may hinder employees’ ability to manage stress effectively. Thus, our study proposes well-being as an underlying mechanism in the associations between organizational and personal resources and burnout levels, suggesting opportunities for organizational interventions during times of crisis.

### 1.1. Theoretical Framework

Several key theoretical frameworks and empirical findings can explain the association between organizational and personal resources, on the one hand, and positive work outcomes, on the other, particularly at times of extra-organizational crisis and life-threatening events, such as terror attacks or wars (i.e., [1,6,8,14]). The conservation of resources (COR) theory [17] offers a crucial lens through which to understand how employees navigate global crises, like wars and a pandemic. COR theory posits that individuals prioritize conserving essential resources—such as time, energy, and emotional stability—to cope with external pressures. Crises like a pandemic, terrorism, and wars severely strain employees’ resources, making it challenging for employees to meet job demands and maintain their well-being [18,19]. This strain heightens stress, reduces coping capacity, and triggers negative psychological outcomes, fueling a negative cycle of resource depletion [20]. This dynamic not only erodes individual well-being, but also exacerbates stress and burnout, with far-reaching effects on organizational performance and resilience. According to the COR theory [17,21,22], individuals are motivated by the desire to acquire, retain, and protect resources that contribute to their well-being. In the midst of war, resource reservoirs are significantly depleted as a result of the threat to life and the risk to the lives of family and friends [8,9,23].

To counter resource loss, individuals should invest their available resources toward the recovery and acquisition of resources. In general, individuals with ample resources are less susceptible to loss and more likely to be able to obtain new resources, whereas individuals with limited resources are more likely to be vulnerable to resource depletion and to have a diminished capacity to acquire new resources [21,22]. Therefore, resilience is primarily determined by the ability of individuals to acquire and maintain stress-resistant resources, which serve as protective mechanisms in the face of external pressures [22]. For instance, Hobfoll et al. [8] found that loss of resources after 9/11 significantly predicted psychological distress, while access to supportive resources, such as social support and counseling, minimized that adverse effect.

Based on the COR theory, the present study examined how organizational and personal resources can contribute to well-being and reduce burnout during a war (Figure 1). It is imperative to understand how resources help employees to cope with external stressors at a time when their environment is fraught with uncertainty and risk. Specifically, we propose that organizational resources, such as perceived organizational support, perceived managerial accessibility, and psychological safety, and personal resources, such as hope and a paradoxical mindset, can improve employee well-being and reduce burnout.

#### 1.1.1. Organizational Resources as Support Factors

Organizational resources are the structural and social support factors that are available to employees in the workplace for the purpose of managing stress and maintaining well-being [24,25,26,27]. These factors include supervisory support, resilience training, and work–life balance practices [24,25,26]. These resources are essential for strengthening resilience, engagement, and job satisfaction and also serve as protective factors against burnout by helping to preserve employees’ physical and emotional resources [18,20,28]. In this study, we examined three critical organizational resources—perceived organizational support, perceived managerial accessibility, and psychological safety—that together constitute a comprehensive framework for coping with stress in challenging situations [24,25,26,27,29].

The term perceived organizational support refers to employees’ perceptions of “the extent to which the organization values their contribution and cares about their well-being” [30] (p. 501). Perceived organizational support plays a crucial role in reducing the adverse effects of high workplace demands by facilitating resilience and adaptive responses, while offering a protective buffer against stress and trauma [18,31,32]. Several systematic reviews [33,34] and meta-analyses [27,35] have demonstrated that perceived organizational support has negative effects on burnout, stress levels, absenteeism, and turnover. During times of war, organizational support is crucial, as employees are exposed to heightened levels of uncertainty and trauma, resulting in the rapid depletion of vital resources. By directly supporting the preservation and recovery of social and psychological resources [7], perceived organizational support helps to shield employees against resource-loss spirals, enhancing resilience and well-being under extreme conditions.

Perceived managerial accessibility is another significant resource, particularly during times of crisis and high stress [1,36,37]. It reflects the extent to which supervisors are available and approachable for communication [38]. As a direct and tangible form of support, managerial accessibility becomes especially vital during disruptive events [25]. Evidence shows that it can buffer the negative effects of workplace stressors and promote employee well-being [25,39].

While perceived organizational support and perceived managerial accessibility are related, they are distinct constructs, with employees able to differentiate between support from their organization and support from their immediate supervisors [25,40,41,42]. For employees to be able to manage stress effectively, both organizational support and managerial accessibility are required.

In accordance with the COR theory, contextual resources play a pivotal role in reducing resource loss [40,42,43]. By providing resources that help employees to handle external stressors and foster a sense of security, organizations can initiate resource-gain spirals—positive cycles in which the acquisition of resources leads to further resource accumulation, ultimately promoting employee well-being and reducing burnout levels [44].

Based on the information presented above, we formulated the following hypotheses (Figure 1):

**Hypothesis** **1a:**
*In the context of an external crisis, perceived organizational support is positively associated with employee well-being and negatively associated with burnout.*


**Hypothesis** **1b:**
*In the context of an external crisis, perceived managerial accessibility is positively associated with employee well-being and negatively associated with burnout.*


According to Edmondson [45], psychological safety is the perception that a workplace is a safe place to express ideas and concerns, enabling employees to express themselves freely and learn without fear of adverse consequences [45]. Psychological safety can be considered an essential resource within the COR framework. Psychological safety creates an environment in which individuals feel comfortable communicating, taking risks, and seeking support. In this way, key resources such as emotional resilience, mental energy, and self-esteem are conserved, thereby reducing the need for self-protective behaviors. Psychological safety also facilitates access to additional resources, such as team support and managerial guidance, creating a resource-gain spiral. Employees who perceive high levels of psychological safety are more likely to engage in behaviors that promote their well-being and to cultivate positive interpersonal relationships [46]. This dynamic strengthens resilience and serves as a buffer against burnout by enabling the accumulation of additional resources over time [47].

Psychological safety serves as a stabilizing resource during crisis situations, as it prevents the rapid depletion of employees’ emotional and cognitive resources by providing a supportive environment in which fears and uncertainties can be expressed. The concept of psychological safety contributes to COR theory’s principle of resource reinforcement, according to which access to one resource strengthens the resilience of others, making it an essential asset when navigating crises [20]. Psychological safety facilitates the safeguarding and retention of other valuable resources for employees [48].

We, therefore, hypothesize the following:

**Hypothesis** **1c:**
*In the context of an external crisis, psychological safety is positively associated with employee well-being and negatively associated with burnout.*


#### 1.1.2. Personal Resources as Support Factors

According to the COR theory, personal resources are attributes and characteristics that enable individuals to cope with stress and adversity, maintain their well-being, and protect, sustain, and build additional resources [17,47]. We examined two personal resources that are essential for coping with external stressors during wartime: hope and paradoxical mindset (Figure 1).

Hope is a psychological resource that encompasses both agency (or the motivation to pursue goals) and pathways (or the strategies to achieve those goals; [49]). Individuals who are hopeful are better able to identify effective paths to success and to sustain their motivation under adverse conditions. Hope provides individuals with vital psychological resources to help them overcome difficulties and thrive [50]; it enhances adaptive coping mechanisms and resilience [51]. In addition, the cognitive ability to self-regulate [52] enables individuals to perform effectively under highly stressful conditions [53]. This aligns with the concept of resource caravans, which posits that individuals possessing key psychological resources, such as hope, have a greater likelihood of attracting and accumulating additional resources over time [54,55,56], thereby increasing their own well-being and reducing burnout [57].

We, therefore, hypothesize the following:

**Hypothesis** **2a:**
*In the context of an external crisis, hope is positively associated with employee well-being and negatively associated with burnout.*


Paradox mindset can be defined as the ability to accept and navigate competing demands and contradictions [58]. It refers to the extent to which individuals are accepting of and even energized by tensions, rather than being hindered by them [58]. Such individuals value and feel comfortable with tensions, viewing them as opportunities, confronting them directly, and seeking both/and strategies instead of either/or solutions [59]. Having this mindset allows individuals to cope more effectively with ambiguity, uncertainty, and resource scarcity. Thus, paradox mindset is a valuable resource, especially during challenging times such as wartime. Using tensions as opportunities, employees with a paradox mindset are likely to experience reduced stress and anxiety as a result of embracing and proactively managing challenges. Paradox mindset fosters both psychological resilience and work engagement [60,61,62].

Paradox mindset can be a valuable resource during wartime, as it enables employees to balance competing demands, such as daily work responsibilities and the uncertainties and dangers of ongoing conflict, without becoming overwhelmed. In accordance with COR theory, an individual with a paradox mindset is able to manage resource loss effectively and employ strategies to mitigate tension [63]. As a result, this mindset has a stress-buffering and resource-boosting effect, which may reduce burnout and improve well-being.

Considering the arguments presented above, we formulated the following hypothesis:

**Hypothesis** **2b:**
*In the context of an external crisis, paradox mindset is positively associated with employee well-being and negatively associated with burnout.*


#### 1.1.3. Well-Being as a Mediator of the Relationship Between Resources and Burnout

To understand the impact of workplace stressors on employees, it is crucial to understand the relationship between well-being and burnout. Seligman [51] defines well-being as a multifaceted construct that encompasses five core dimensions: positive emotions, engagement, relationships, meaning, and accomplishment. This framework emphasizes the importance of looking at well-being in a holistic manner across many aspects of one’s life. In contrast, the term burnout refers to a prolonged response to chronic emotional and interpersonal stressors on the job that may occur due to external and internal factors. Burnout is characterized by the feeling of exhaustion, depersonalization, and a reduced sense of accomplishment [64].

Although burnout and well-being are closely related, empirical evidence indicates that they are distinct concepts (e.g., [64,65,66,67,68]). While little research has been conducted on the relationship between employee well-being and burnout [69], there is some evidence to suggest that well-being is an antecedent of burnout levels. For instance, research has shown that employees with higher levels of well-being are better equipped to utilize their personal resources and optimize the benefits of organizational resources, which helps to mitigate burnout (e.g., [20,47]).

COR theory [17] provides a framework for understanding the interplay between well-being, personal and organizational resources, and burnout. According to COR theory, the availability of resources plays a critical role in maintaining well-being and preventing resource depletion. Personal and organizational resources help to foster well-being, which acts as a buffer against stress and mitigates the risk of burnout. Conversely, a lack of resources initiates a resource-loss spiral, in which diminished well-being leads to further resource depletion and increased susceptibility to burnout [70]. Personal resources, such as hope and a paradox mindset, enhance well-being by equipping individuals with the psychological tools needed to manage stress effectively and maintain resilience [49,56]. Organizational resources, including perceived organizational support and psychological safety, play a complementary role by providing an environment that replenishes resources and reduces stress [30,45]. Together, these resources enhance employees’ psychological well-being, enabling them to cope with workplace demands more effectively and reducing the risk of burnout.

Based on the above, we formulated several hypotheses about organizational resources (Hypothesis 3a–c) and personal resources (Hypothesis 4a,b).

**Hypothesis** **3a:**
*Well-being mediates the relationship between positive organizational support and employee burnout.*


**Hypothesis** **3b:**
*Well-being mediates the relationship between positive managerial accessibility and employee burnout.*


**Hypothesis** **3c:**
*Well-being mediates the relationship between psychological safety and employee burnout.*


**Hypothesis** **4a:**
*Well-being mediates the relationship between hope and employee burnout.*


**Hypothesis** **4b:**
*Well-being mediates the relationship between paradox mindset and employee burnout.*


## 2. Methods

### 2.1. The Research Context

In terms of its magnitude, the 7 October 2023 attack in southern Israel was unprecedented. Based on the number of victims per capita, this attack ranks as the deadliest terrorist attack in recent decades [71,72,73]. In the aftermath of the terror attack, many Israelis experienced direct losses of family members and friends, while others experienced life-threatening events during missile attacks and significant concerns regarding family members who were immediately called up for military service. In a nationally representative sample of Israeli citizens, the prevalence of probable post-traumatic stress disorder almost doubled between a few weeks before the attack and 6 weeks after the attack [74]. A total of 150,000 citizens were evacuated from their homes, due to their proximity to the northern and southern borders of Israel [75]. While violence remained a threat, economic life continued in Israel and organizations, factories, and kibbutzim throughout the country continued to operate. All of this served as a starting point for our research. Life losses and exposure to traumatic events occurred on the other side of the border as well, but we were unable to collect data on the Palestinian side.

### 2.2. Sample and Data-Collection Procedures

Participants were recruited via an online survey firm with access to a diverse pool of employees across various occupations and work roles. Data were collected only from respondents on the Israeli side of the border, as access to participants on the Palestinian side was not feasible. All participants received a small honorarium for their participation. The eligibility criteria included salaried employees who had worked for their current employer for more than 6 months. The study protocol was approved by the Institutional Review Board.

The first wave of data collection (T1) took place during the early days of the war in Israel–3 weeks after the attack of 7 October 2023. At that time, participants completed an online survey that assessed their demographic characteristics, direct exposure and proximity to the attack, organizational resources (i.e., organizational support, psychological safety, managerial accessibility), and personal resources (i.e., hope, paradox mindset). The second wave of data collection (T2) occurred a month later, as the country continued to cope with the ongoing conflict. At T2, the same participants were asked to complete a second survey, which included well-being scale measured by the workplace PERMA profiler [76] and a measure of burnout [77]. The one-month time lag was employed to reduce potential common method variance [78] and to allow for a more robust examination of directional relationships between predictors and outcomes.

A total of 430 individuals responded to the T1 survey and 304 followed up with the T2 survey (retention rate = 70.6%). Among the individuals who participated in both waves of data collection, the average age was 42.38 years (*SD* = 11.93), 53.9% of the participants were women, and 6.9% reported having been evacuated from their home.

### 2.3. Measures

To ensure the reliability of the translation process, all study scales were translated from English into Hebrew and then back-translated into English.

#### 2.3.1. Organizational Resources

The following organizational resources were assessed at T1: perceived organizational support, perceived managerial accessibility, and psychological safety. Perceived organizational support was measured using Mihalache and Mihalache’s (Amsterdam, The Netherland) [25] six-item Survey of Perceived Organizational Support. (Sample item: “My company provides satisfactory measures for supporting communication with colleagues working at different locations;” α = 0.94).

Perceived managerial accessibility was evaluated using Mihalache and Mihalache’s [25] four-item Survey of Supervisor Accessibility. (Sample item: “When needed, my manager takes time for me;” α = 0.93.) Responses for both organizational support and managerial accessibility were scored on a 5-point Likert scale ranging from 1 (strongly disagree) to 5 (strongly agree).

Psychological safety was assessed using Edmondson’s (Ithaca, NY, USA) [45] seven-item Psychological Safety Scale. (Sample item: “If you make a mistake, it is often held against you;” reverse scored; α = 0.65.) Participants rated their agreement using a 7-point scale ranging from 1 (strongly disagree) to 7 (strongly agree).

#### 2.3.2. Personal Resources

The following personal resources were also measured at T1: hope and paradox mindset. Hope was evaluated using Snyder et al.’s (Lawrence, Kansas) [49] 12-item Adult Hope Scale, which encompasses four items measuring pathways thinking, four items assessing agency thinking, and four filter items. The total score is derived from the sum of the pathway and agency items, with higher scores indicating greater levels of hope. (Sample item: “I can think of many ways to get out of a jam;” α = 0.84.) Participants responded on a scale ranging from 1 (definitely false) to 7 (definitely true).

Paradox mindset was measured using Miron-Spektor et al.’s (Fontainebleau, France) [58] nine-item scale. (Sample item: “When I consider conflicting perspectives, I gain a better understanding of an issue;” α = 0.89.) Participants rated their agreement using a 7-point scale from 1 (strongly disagree) to 7 (strongly agree).

#### 2.3.3. Employee Well-Being and Burnout

Employee well-being and burnout were measured at T2. Employee well-being was evaluated using the Workplace PERMA Profiler (Melbourne, Australia) [76,79], which is a 16-item scale designed to measure the components of Seligman’s [51] flourishing model across five dimensions (i.e., positive emotion, engagement, relationships, meaning, and accomplishment) in the workplace context. The scale consists of three items for each dimension, along with one item assessing global happiness. (Sample item: “At work, how often do you feel joyful?” α = 0.95.) Participants rated their agreement on a scale ranging from 0 (not at all/never) to 10 (completely/always).

Burnout was assessed using Malach-Pines’s (Beer Sheva, Israel) [77] 10-item Burnout Measure Short Version (BMS). (Sample item: “I feel exhausted;” α = 0.91). Participants indicated their level of agreement using a scale ranging from 1 (never) to 7 (always).

#### 2.3.4. Control Variables

We controlled for employees’ age and gender, as previous research has established associations between these demographics and burnout [80]. We also controlled for two indicators of exposure to war. The first indicator was the time available to reach a shelter following a warning of an attack (e.g., air-raid siren) based on the participant’s location; shorter times indicate greater proximity to danger. This variable was measured on a scale ranging from 1 (15 s) to 4 (90 s). The second indicator captured personal proximity via social ties and was measured with a single item: “To what extent did you have a personal connection to someone who was injured or killed on October 7 or in the ongoing conflict?” Responses were recorded on a 5-point scale (1 = not at all; 2 = slightly; 3 = moderately; 4 = quite a lot; 5 = to a very high degree). To ensure consistent interpretation, participants were instructed to select the level reflecting their closest relationship to any such person (e.g., distant acquaintance/co-worker/neighbor; friend/extended family; close friend/close relative; immediate family member/partner). Controlling for these variables is crucial, as exposure to traumatic events can significantly affect both well-being scores and levels of burnout.

## 3. Results

An overview of the descriptive statistics and the correlations between the study variables is presented in Table 1. To address potential concerns regarding common method variance [78], a confirmatory factor analysis (CFA) was conducted using a nine-factor model. This model included eight factors representing the study variables (i.e., organizational support, managerial accessibility, psychological safety, hope, paradox mindset, well-being, and burnout), as well as an additional method factor on which all items were loaded. The results indicated an acceptable model fit: χ^2^(1647) = 2806.52, *p* < 0.01; CFI = 0.91; TLI = 0.90; SRMR = 0.06; RMSEA = 0.05. Importantly, the common method factor accounted for only 5.76% of the variance, which is below the recommended 50% threshold [78].

To test Hypotheses 1–4, we conducted a path analysis using AMOS 19 [81], controlling for age, gender, physical proximity to the war zone (i.e., time available to seek shelter), and extent of personal connection with individuals injured or killed on October 7th or during the subsequent war. A two-step approach was employed [82]. The first step involved assessing the measurement model and evaluating construct independence through CFA. In the second step, we tested the research model. The CFA results for the proposed seven-factor model demonstrated an acceptable fit to the data (χ^2^ = 2784.95; *df* = 1647; *p* < 0.001; CFI = 0.91; TLI = 0.90; SRMR = 0.06; RMSEA = 0.05). Furthermore, the seven-factor hypothesized model fit the data far better than alternative models (see Table 2).

For the path analysis, we specified the relationship between organizational support, psychological safety, managerial accessibility, paradox mindset, hope, well-being, and burnout. Following Becker’s [83] recommendation, we included only the control variable relationships that were significantly correlated with the dependent variables. To estimate the indirect effects, we utilized a bias-corrected bootstrap procedure [84] with 10,000 replications. The model demonstrated an acceptable fit to the data (χ^2^ = 62.55; *df* = 32; *p* = 0.001; CFI = 0.95; TLI = 0.92; SRMR = 0.07; RMSEA = 0.06). As hypothesized, organizational resources variables were positively associated with well-being scores (β = 0.42, *SE* = 0.10, *p* < 0.001; β = 0.30, *SE* = 0.11, *p* = 0.004; β = 0.39, *SE* = 0.10, *p* < 0.001 for organizational support, psychological safety, and managerial accessibility, respectively) and well-being scores were negatively associated with employee burnout (β = −0.24, *SE* = 0.03, *p* < 0.001). The indirect relationships between the different organizational resources and burnout via well-being scores were significant [indirect effect = −0.09, *SE* = *0*.03, 95% bias-corrected confidence interval BCCI (−0.16, −0.04) for organizational support; indirect effect = −0.07, *SE*= 0.03, 95% BCCI (−0.14, −0.02) for psychological safety; indirect effect = −0.08, *SE*= 0.03, 95% BCCI (−0.15, −0.04) for managerial accessibility], confirming Hypothesis 1a–c.

Regarding personal resources, the results showed that while the association between a paradox mindset and well-being scores was not significant (β = 0.02, *SE* = 0.09, *ns*), hope was positively associated with well-being scores (β = 0.38, *SE* = 0.09, *p* < 0.001). The indirect relationship between hope and burnout, mediated by well-being scores, was also significant [indirect effect = −0.08, *SE* = 0.03, 95% BCCI (−0.15, −0.04)]. These findings offer partial support for the overall research model (see Figure 2).

To evaluate the influence of statistical controls on our findings [83], we re-ran the analyses without including control variables. The results remained consistent with the original model, demonstrating an acceptable fit (χ^2^ = 6.20, *df* = 3, *ns*; CFI = 0.99; TLI = 0.96; SRMR = 0.04; RMSEA = 0.06). The indirect effects of perceived organizational support [indirect effect = −0.09, *SE* = 0.03, 95% BCCI (−0.17, −0.04)], psychological safety [indirect effect = −0.07, *SE* = 0.03, 95% BCCI (−0.14, −0.02)], managerial accessibility [indirect effect = −0.09, *SE* = 0.03, 95% BCCI (−0.15, −0.04)], and hope [indirect effect = −0.08, *SE* = 0.03, 95% BCCI (−0.15, −0.04)] also remained stable. These findings confirm that the inclusion of the control variables in the model did not significantly alter the overall pattern of the results.

## 4. Discussion

This study advances our understanding of how personal and organizational resources contribute to employee well-being and mitigate burnout during wartime. The findings extend the COR theory [17] by highlighting the mediating role of well-being in the relationship between resources and burnout, offering a robust mechanism for resource conservation under conditions of prolonged stress and uncertainty. These findings demonstrate that organizational resources, such as perceived organizational support, perceived managerial accessibility, and psychological safety, significantly contribute to employee well-being. These findings align with prior research, which has underscored the buffering effects of organizational resources in mitigating stress and preserving employees’ psychological resources [30,35].

In addition, personal resources, particularly hope, were found to positively influence well-being, demonstrating the protective role that inherent psychological assets play in mitigating the effects of resource loss [49,56]. Employees with higher levels of hope are better equipped to identify pathways to overcome challenges, maintain motivation, and build resilience, all of which reduce their risk of burnout. This finding emphasizes the protective function of hope, which becomes particularly salient in the context of crises such as war [85].

Interestingly, the hypothesized role of paradox mindset was not supported. Although paradox mindset has been shown to facilitate adaptability and reduce stress in routine workplace environments, its effectiveness during extreme crises, such as war, appears limited. One possible explanation is that the benefits of a paradox mindset may become more apparent over prolonged periods of stress. As crises evolve, the ability to balance conflicting demands and frame contradictions as opportunities may help individuals to conserve resources and engage in small, positive actions that enhance well-being. However, during acute phases of existential threats and heightened emotional strain, paradox mindset may lack the immediate utility needed to counterbalance resource depletion. Moreover, paradox mindsets may operate differently depending on the individual’s baseline resilience or coping mechanisms. It may be difficult for employees who are already depleted of resources or who are facing acute personal challenges to balance competing demands, suggesting that a paradox mindset may be of limited benefit in the absence of adequate resource reserves.

In summary, our findings underscore the critical role of well-being as a central mechanism linking resources with burnout. Organizational resources provide a supportive framework for employees, enabling them to preserve and replenish psychological resources. Simultaneously, personal resources, such as hope, fortify individuals’ capacity to navigate stress and maintain resilience. Together, these resources mitigate burnout by fostering well-being, which acts as a buffer against the cascading effects of prolonged stress and uncertainty.

### 4.1. Theoretical and Practical Contributions

This study makes significant theoretical contributions by advancing our understanding of the interplay between personal and organizational resources in shaping employee outcomes during crises [9,13]. The findings extend the COR theory by illustrating that well-being has a dual role in resource conservation and replenishment in high-stress contexts. Specifically, the results demonstrate that well-being acts as a psychological buffer, enabling individuals to maintain their resource reserves while mitigating the effects of resource depletion. The findings of this study are consistent with previous research, which has emphasized the cyclical nature of resource loss and gain, demonstrating that well-being can catalyze resource-gain spirals, particularly when supported by robust support systems [19,70].

Our findings also contribute to the discussion regarding the distinct nature of well-being and burnout as separate, but related constructs [64,65,66,67,68,70]. As previously demonstrated [28,47], high levels of well-being reduce the risk of burnout by replenishing emotional and cognitive resources. However, interventions designed to prevent burnout must also address its unique predictors, including excessive tension and a lack of recovery opportunities [18].

This study highlights the critical importance of organizational strategies for fostering resilience and well-being during times of crisis. Our findings underscore the necessity of creating environments that provide robust psychological safety and managerial accessibility, as these organizational resources are essential for preserving psychological reserves and facilitating stress recovery [30,45]. To enhance employee resilience during crises, organizations should adopt actionable strategies that address both individual and organizational needs.

The following are some practical recommendations. First, fostering supportive organizational and managerial practices—such as open communication, empathy, and personalized resource allocation—can help employees to feel valued and understood, reducing stress and promoting engagement [30,86]. Second, cultivating psychological safety is essential. By creating environments in which employees feel comfortable to voice concerns and to take risks without fear of negative consequences, organizations can encourage adaptability and innovation even under challenging circumstances [46]. Due to the prolonged exposure to threat, all these support systems provide employees with an opportunity to openly discuss and express their fears, stress, and anxiety. Third, promoting employee hope can empower individuals to navigate uncertainty with optimism and cognitive flexibility. Hope can be nurtured and cultivated through positive interventions [87]. Resilience training programs that focus on personal resources, such as hope, may enhance employees’ intrinsic abilities to cope with adversity. Such interventions can amplify resource-gain spirals, enabling employees to use their intrinsic strengths to mitigate stress and maintain performance [55].

The limited role played by the paradox mindset in this study suggests that interventions targeting cognitive flexibility must be carefully tailored to the unique demands of extreme crisis situations. Finally, well-being as measured by the Workplace PERMA Profiler [76,79] offers a valuable framework for designing tailored interventions by targeting multiple facets of well-being—such as positive emotions, engagement, and relationships, ensuring a holistic approach to employee support. By integrating these strategies, organizations can build a more resilient workforce capable of thriving in the face of crises.

### 4.2. Limitations and Future Directions

This study is not without its limitations. The reliance on self-reported measures may introduce response biases and the cultural and geopolitical context of the study limits the generalizability of its findings. Future research should consider longitudinal designs and cross-cultural comparisons to validate these findings in diverse settings. Additionally, further exploration of the conditions under which paradoxical thinking contributes to well-being and burnout mitigation is warranted. For example, future studies might investigate how variables such as time, context, and individual differences affect the efficacy of paradoxical thinking in crisis scenarios. Further examination of the interplay between personal traits, environmental factors, and crisis severity could provide deeper insights into the nuanced roles of different personal resources.

Lastly, while this study offers important insights into the experiences of Israeli civilians during a time of war, it does not address the perspectives or hardships faced by Palestinian civilians. This represents a significant limitation in capturing the broader psychological consequences of the ongoing conflict. Future research should strive to adopt a more inclusive approach that considers the impact on all affected populations.

### 4.3. Conclusions

In conclusion, this study underscores the critical importance of integrating personal and organizational resources to safeguard employee well-being and prevent burnout during a crisis. By highlighting the mediating role of well-being and identifying the differential impacts of personal and organizational resources, these findings offer valuable guidance for organizations aiming to enhance resilience and support their workforce in navigating prolonged adversity.

## Figures and Tables

**Figure 1 ijerph-22-01384-f001:**
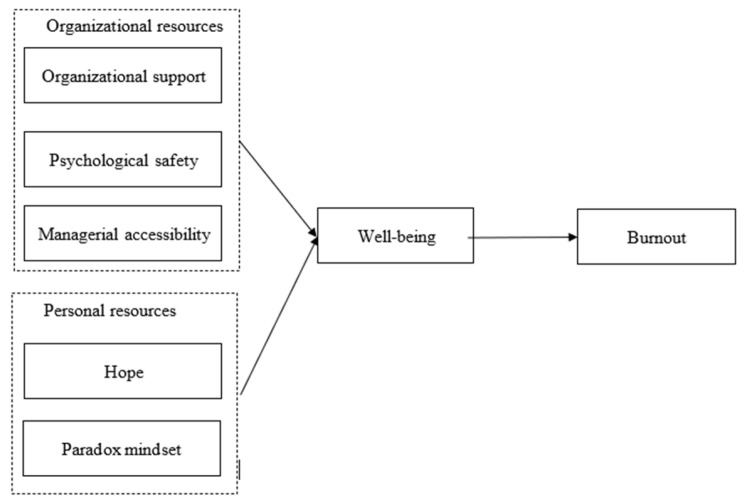
Research Model.

**Figure 2 ijerph-22-01384-f002:**
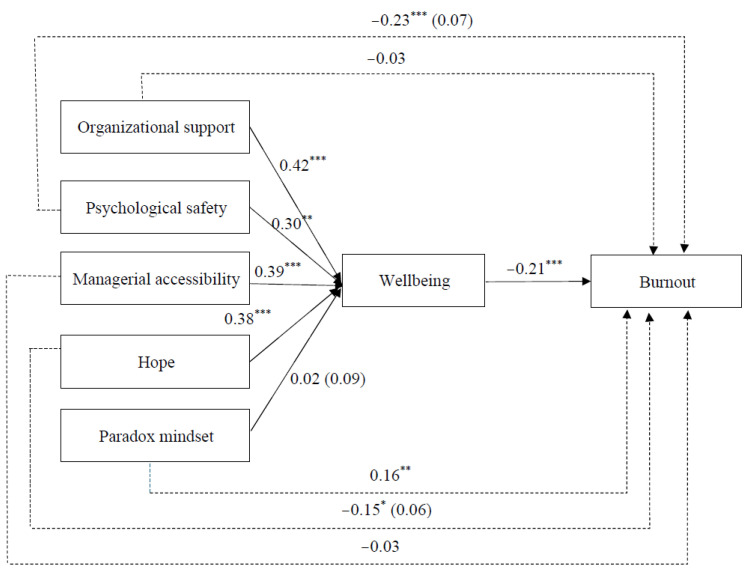
Results of the Path-Analysis. Note. Unstandardized coefficients are shown, with standard errors in parentheses. * *p* < 0.05. ** *p* < 0.01. *** *p* < 0.001.

**Table 1 ijerph-22-01384-t001:** Means, Standard Deviations, and Correlations Among the Study Variables.

	*M* (*SD*)	1	2	3	4	5	6	7	8	9	10
1. Gender ^a^	0.46 (0.50)										
2. Age	42.38 (11.93)	−0.01									
3. Proximity to war zone	3.26 (0.97)	−0.02	−0.03								
4. Attack exposure	2.55 (1.23)	−0.02	−0.05	−0.05							
5. Organizational support	3.43 (1.08)	−0.02	0.13 *	0.05	−0.07						
6. Psychological safety	4.59 (0.91)	−0.09	0.04	0.14 *	0.05	0.35 ***					
7. Managerial accessibility	3.74 (1.07)	−0.12 *	−0.00	0.08	−0.03	0.62 ***	0.42 ***				
8. Hope	6.09 (1.06)	0.01	−0.05	0.06	0.18 **	0.24 ***	0.40 ***	0.25 ***			
9. Paradox mindset	4.14 (1.03)	0.17 **	−0.04	0.06	0.16 **	0.12 *	0.10	0.05	0.42 ***		
10. Well-being	7.78 (1.85)	−0.06	0.09	0.12 *	0.01	0.50 ***	0.43 ***	0.50 ***	0.40 ***	0.16 **	
11. Burnout	3.30 (1.16)	−0.15 **	−0.15 *	−0.08	0.22 ***	−0.31 ***	−0.36 ***	−0.30 ***	−0.26 ***	0.01	−0.47 ***

Note. *N* = 304. ^a^ 0 = female; 1 = male. * *p* < 0.05. ** *p* < 0.01. *** *p* < 0.001.

**Table 2 ijerph-22-01384-t002:** Confirmatory Factor Analyses.

Model	χ^2^	*df*	CFI	TLI	SRMR	RMSEA	Δχ^2^(Δ *df*) ^a^
Intended model ^b^	2784.95	1647	0.91	0.90	0.06	0.05	----
1-factor model ^c^	6773.00	1674	0.59	0.56	0.12	0.10	3988.05 **
2-factor model ^d^	5792.52	1673	0.67	0.65	0.12	0.09	3007.57 **
3-factor model ^e^	4967.57	1672	0.73	0.72	0.11	0.08	2182.62 **

Note. ** *p* < 0.01. ^a^ Comparison with the intended 7-factor model. ^b^ Hypothesized 7-factor model. ^c^ One-factor model-general model; all study variables were loaded on one factor. ^d^ Two-factor model in which items measured at T1 and T2 were loaded on two different factors. ^e^ Three-factor model: (f1) organizational resource–organizational support, managerial accessibility, and psychological safety; (f2) personal resource–hope and paradox mindset; (f3) dependent variables–PERMA and burnout.

## Data Availability

Data is available from the corresponding author upon reasonable request.

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
