# Peer review of "Double Shield: The Roles of Personal and Organizational Resources in Promoting Positive Outcomes for Employees During Wartime"

_ijerph, 2025, doi:10.3390/ijerph22091384_

Round 1

Reviewer 1 Report

Comments and Suggestions for Authors

Well-researched and written article. Under the abstract, kindly include the type of data collected and how data were analysed.

Author Response

Reviewer 1

Dear Reviewer,

We would like to thank you for your careful reading of our manuscript and for the constructive comments and suggestions. Your feedback has been very helpful in clarifying our presentation and strengthening the manuscript. Below, we address each of your points in detail and describe the changes we have made accordingly. For ease of reference, we have quoted your comments and provided our responses in red font beneath each one.

  1. Absract - Give the reader an idea on how the type of data collected and how data was analysed.

Response: We thank the reviewer for this helpful comment. In the revised abstract, we now clarify that data were collected through a two-wave online survey administered by a professional survey firm, providing access to a diverse pool of Israeli employees across occupations and work roles in November and December 2023, following the October 7 terrorist attack by Hamas. We further specify that data were analyzed using path analysis with bootstrapped indirect effects (please see Abstract, p.1).

  1. Abstract - Please check to see if the journal requires this in alphabetical order

Response: As suggested, we have revised the keywords in the abstract to appear in alphabetical order.

  1. 7 - and then back translated to which language?

Response: We appreciate the reviewer’s careful attention. We have revised the manuscript accordingly, clarifying that “to ensure the reliability of the translation process, all study scales were translated from English into Hebrew and then back-translated into English” (p. 7).

Reviewer 2 Report

Comments and Suggestions for Authors

The authors conducted a meaningful study. The quality of the manuscript is quite good. Below you will find few comments. I hope that revision and correction of the manuscript will improve its quality.

1) The authors discuss the outcomes of Paradox mindset, but does not provide a definition of this phenomenon in the Theoretical Framework. Every concept of the study should be clearly defined.

2) several assumptions and hypotheses were formulated incorrectly considering that a cross sectional but not longitudinal or experimental research design was used in the study:.

a) "we test whether reduced well-being can exacerbate burnout because it depletes the personal resources required to manage stress effectively"

b) Hypothesis 1a: During external crises, perceived organizational support is positively associated with employee well-being and negatively associated with burnout. Also, other hypotheses in which are written "During external crises"

3) I disagree that Supervisor Accessibility is the same like Managerial Support. The support is wider than accessibility. I recommend to the authors to think very carefully upon these concepts.

4) I suggest to use only one concept Personal or Individual (see chapter 2.3.2) Resources, but not both.

5) Please, describe more clearly this variable "The second indicator pertained to personal connections with individuals who were injured or killed during the incident on October 7th or in the ongoing conflict, measured on a scale from 1 (not at all) to 5 (to a very high degree). I did not understand what is evaluated in the scale from 1 to 5.

6) I disagree with the authors that their research "supports the notion that well-being and burnout are distinct psychological states that do not represent two extremes on a continuum". The correlation which was found between well-being and burnout may support the idea that these two phenomena are on a continuum. 

Author Response

Reviewer 2

The authors conducted a meaningful study. The quality of the manuscript is quite good. Below you will find few comments. I hope that revision and correction of the manuscript will improve its quality.

We would like to thank you for your careful reading of our manuscript and for the constructive comments and suggestions. Your feedback has been very helpful in clarifying our presentation and strengthening the manuscript. Below, we address each of your points in detail and describe the changes we have made accordingly. For ease of reference, we have quoted your comments and provided our responses in red font beneath each one.

  1. The authors discuss the outcomes of Paradox mindset, but does not provide a definition of this phenomenon in the Theoretical Framework. Every concept of the study should be clearly defined.

Response: We thank the reviewer for this helpful comment. In line with the suggestion, we added a clear definition of paradox mindset in the Theoretical Framework section. Specifically, we now state: Paradox mindset can be defined as the ability to accept and navigate competing demands and contradictions [58]. It refers to the extent to which individuals are accepting of and even energized by tensions, rather than being hindered by them [58]. Such individuals value and feel comfortable with tensions, viewing them as opportunities, confronting them directly, and seeking both/and strategies instead of either/or solutions (Lewis, 2000)” (please see p. 6).

Lewis, M. W. (2000). Exploring paradox: Toward a more comprehensive guide. Academy of Management review25(4), 760-776.

Miron-Spektor, E., Ingram, A., Keller, J., Smith, W. K., & Lewis, M. W. (2018). Microfoundations of organizational paradox: The problem is how we think about the problem. The Academy of Management Journal61(1), 26-45.

2) several assumptions and hypotheses were formulated incorrectly considering that a cross sectional but not longitudinal or experimental research design was used in the study:

  1. a) "we test whether reduced well-being can exacerbate burnout because it depletes the personal resources required to manage stress effectively"

Response: We thank the reviewer for this important comment. We agree that the original wording implied a stronger causal claim than is appropriate for our survey design. To address this, we revised the statement to remove causal language and emphasize associations. The revised wording now reads: “Specifically, we examine whether lower levels of well-being are associated with higher levels of burnout, consistent with the idea that diminished personal resources may hinder employees’ ability to manage stress effectively. “(please see p.2).

  1. b) Hypothesis 1a: During external crises, perceived organizational support is positively associated with employee well-being and negatively associated with burnout. Also, other hypotheses in which are written "During external crises"
  2. b) Hypothesis 1a: During external crises, perceived organizational support is positively associated with employee well-being and negatively associated with burnout. Also, other hypotheses in which are written "During external crises"

Response: We thank the reviewer for this observation. We agree that the original phrasing “During external crises” could imply a causal or longitudinal assumption beyond the scope of our time-lagged survey design. To address this, we revised all hypotheses to describe the crisis context without suggesting causality or longitudinal effects. For example, Hypothesis 1a now reads: In the context of an external crisis, perceived organizational support is positively associated with employee well-being and negatively associated with burnout.”  And Hypothesis 1b reads: “In the context of an external crisis, perceived managerial accessibility is positively associated with employee well-being and negatively associated with burnout.” (please see p. 5). We applied this revised wording consistently across all hypotheses that previously began with “During external crises”.

3) I disagree that Supervisor Accessibility is the same like Managerial Support. The support is wider than accessibility. I recommend to the authors to think very carefully upon these concepts.

Response: We appreciate this important clarification and agree that managerial support is broader than managerial accessibility. To avoid conceptual conflation, we standardized the terminology to “managerial accessibility” throughout the manuscript and distinguished it explicitly from the broader notion of managerial support. We also revised the hypotheses, tables, and figure labels to use “managerial accessibility.” In the Theoretical Framework, we now refer to this construct: “Perceived managerial accessibility is another significant resource, particularly during times of crisis and high stress [1, 36, 37]. It reflects the extent to which supervisors are available and approachable for communication [Atuahene-Gima & Li, 2002]. As a direct and tangible form of support, managerial accessibility becomes especially vital during disruptive events [Mihalache & Mihalache, 2022]. Evidence shows that it can buffer the negative effects of workplace stressors and promote employee well-being [Mihalache & Mihalache, 2022; Andersen, Pihl-Thingvad, & Andersen, 2025].” (Please see p. 4).

4) I suggest to use only one concept Personal or Individual (see chapter 2.3.2) Resources, but not both.

Response: We thank the reviewer for this important suggestion. We fully agree that using both personal resources and individual resources interchangeably created inconsistency and potential confusion. To address this, we have standardized the terminology to “personal resources” throughout the manuscript.

5) Please, describe more clearly this variable "The second indicator pertained to personal connections with individuals who were injured or killed during the incident on October 7th or in the ongoing conflict, measured on a scale from 1 (not at all) to 5 (to a very high degree). I did not understand what is evaluated in the scale from 1 to 5.

Response: We thank the reviewer for pointing out the lack of clarity regarding this variable. We identified a mistake in the original description of the scale and have now corrected it in the manuscript. The item assesses the degree of personal connection (closeness) to anyone injured or killed on October 7 or in the ongoing conflict. The revised Methods text now reads (p. 9): The original survey question asked: “To what extent did you have a personal connection to someone who was injured or killed on October 7 or in the ongoing conflict?Responses were given on a 5-point scale (1 = not at all; 2 = slightly; 3 = moderately; 4 = quite a lot; 5 = to a very high degree), with higher scores indicating closer personal ties. To ensure consistent interpretation, participants were instructed to select the level reflecting their closest relationship to any such person (e.g., distant acquaintance/co-worker/neighbor; friend/extended family; close friend/close relative; immediate family member/partner). Please see p. 9.

6) I disagree with the authors that their research "supports the notion that well-being and burnout are distinct psychological states that do not represent two extremes on a continuum". The correlation which was found between well-being and burnout may support the idea that these two phenomena are on a continuum. 

Response: We appreciate the reviewer’s thoughtful observation and have revised the manuscript accordingly. Specifically, we have removed this sentence from the Discussion to avoid overstating our conclusions. However, it is important to note that the observed correlation (r = .47) is commonly interpreted as moderate rather than high (Cohen, 1988) and is well below levels typically taken to indicate construct redundancy (latent r ≳ .85; Shaffer, DeGeest, & Li, 2016; van Mierlo, Vermunt, & Rutte, 2009). These results therefore support the treatment of well-being and burnout as related yet non-interchangeable constructs.

References

Cohen, J. (1988). Statistical Power Analysis for the Behavioral Sciences (2nd ed.). Lawrence Erlbaum Associates.
Shaffer, J. A., DeGeest, D., & Li, A. (2016). Tackling the problem of construct proliferation: A guide to assessing the discriminant validity of conceptually related constructs. Organizational Research Methods, 19(1), 80–110.

van Mierlo, H., Vermunt, J. K., & Rutte, C. G. (2009). Composing group-level constructs from individual-level survey data. Organizational Research Methods, 12, 368-392.

Reviewer 3 Report

Comments and Suggestions for Authors

The study is highly interesting and emphasizes the importance of employee well-being and the relationship with employee burnout during war times (the context). The results points towards strategies for organizations/management to sustain well-being and build up buffers against burnout in a country-wide crisis situation and the authors draw on and discuss the results with COR theory.  

However, few clarifications and improvements are needed:

Overall, good and convincing theoretical framework and convincing rationalization for hypotheses and it is highly important that it is stated/acknowledged in at least two places that the authors were unable to collect data among civilians on the Palestinian side.

It is though important that the term PERMA be replaced with the term Well-being in the model presented (figure 1 page 3) and in tables and figure 2 as only after reading through the whole paper it becomes clear to the reader that well-being is the key theoretical concept and construct measured and used in the study. It does thus not seem important to use the term PERMA specifically in the model even though the origin of the well-being definition used and the scale used need to be explained in the method section and theoretical section. The PERMA abbreviation in the model in the theoretical discussion, method and in figures and tables only causes confusion for the reader regarding the concept of well-being. Well-being is stated as a key concept already in the the opening statement in the abstract and also discussed in introduction as a key construct but then the authors switch somewhat abruptly to PERMA elsewhere.  The abbreviation PERMA thus seems to be used somewhat interchangeably with the concept of well-being throughout the paper which is highly confusing - causing a conceptual confusion for readers that may not be familiar with various different scales nor this specific scales used to measure work related well-being.  

The authors also have to state in the abstract and then explain shortly in the method section why the study is designed as Time lagged (T1 and T2) and why key employee outcome measures of well-being (PERMA) and burnouts are collected a month later.  

Author Response

Reviewer 3

Dear Reviewer,

We would like to thank you for your careful reading of our manuscript and for the constructive comments and suggestions. Your feedback has been very helpful in clarifying our presentation and strengthening the manuscript. Below, we address each of your points in detail and describe the changes we have made accordingly. For ease of reference, we have quoted your comments and provided our responses in red font beneath each one.

Comment 1:

 The study is highly interesting and emphasizes the importance of employee well-being and the relationship with employee burnout during war times (the context). The results points towards strategies for organizations/management to sustain well-being and build up buffers against burnout in a country-wide crisis situation and the authors draw on and discuss the results with COR theory. However, few clarifications and improvements are needed:

We thank the reviewer for the positive and encouraging feedback on our study and its contribution. We address the specific clarifications and improvements point by point below.

  1. Overall, good and convincing theoretical framework and convincing rationalization for hypotheses and it is highly important that it is stated/acknowledged in at least two places that the authors were unable to collect data among civilians on the Palestinian side.

Response: Thank you for this comment. In line with this comment, we now explicitly acknowledge the limitation that data could not be collected from civilians on the Palestinian side in both the Introduction: “…we test our hypothesis based on data obtained from the Israeli side of the border due to a lack of access to respondents on the Palestinian side” (please see p. 2) and the Methods section: “Data were collected from respondents on the Israeli side of the border, as access to participants on the Palestinian side was not feasible.” (please see p. 7).

  1. It is though important that the term PERMA be replaced with the term Well-being in the model presented (figure 1 page 3) and in tables and figure 2 as only after reading through the whole paper it becomes clear to the reader that well-being is the key theoretical concept and construct measured and used in the study. It does thus not seem important to use the term PERMA specifically in the model even though the origin of the well-being definition used and the scale used need to be explained in the method section and theoretical section. The PERMA abbreviation in the model in the theoretical discussion, method and in figures and tables only causes confusion for the reader regarding the concept of well-being. Well-being is stated as a key concept already in the the opening statement in the abstract and also discussed in introduction as a key construct but then the authors switch somewhat abruptly to PERMA elsewhere.  The abbreviation PERMA thus seems to be used somewhat interchangeably with the concept of well-being throughout the paper which is highly confusing - causing a conceptual confusion for readers that may not be familiar with various different scales nor this specific scales used to measure work related well-being.  

Response: We thank the reviewer for this important observation and fully agree that the term well-being should be consistently emphasized as the key theoretical construct. In line with your suggestion, we carefully reviewed the manuscript to ensure consistency. We replaced the abbreviation PERMA with well-being in the theoretical model (Figure 1, page 3), in Figure 2, and in all tables. Please note that changes in the figures could not be tracked using the track-changes system. We also revised the text throughout the manuscript to consistently use well-being as the central construct, while clarifying in the theoretical background and methods sections that the PERMA scale was the instrument used to operationalize well-being in this study. We believe these revisions eliminate potential conceptual confusion and enhance the clarity and accessibility of the paper for readers who may be less familiar with specific measurement scales.

  1. The authors also have to state in the abstract and then explain shortly in the method section why the study is designed as Time lagged (T1 and T2) and why key employee outcome measures of well-being (PERMA) and burnouts are collected a month later.  

Response: We thank the reviewer for this valuable comment. In line with the suggestion, we revised both the abstract and the Methods section to clarify the rationale for the time-lagged design. Specifically, the abstract now reads: “Data were collected through an online two-wave survey administered by a professional survey firm with access to a diverse pool of Israeli employees across occupations and work roles in November (time 1) and December 2023 (time 2), following the October 7 terrorist attack by Hamas. A time-lagged design, with key outcomes collected one month after the predictors, was employed to reduce the risk of common method bias.” (Please see p. 1). In addition, the Methods section was revised to state: “The one-month time lag was employed to reduce potential common method variance [78] and to allow for a more robust examination of directional relationships between predictors and outcomes.” (please see p.8).

  1. Podsakoff, P.M.; MacKenzie, S.B.; Lee, J.Y.; Podsakoff, N.P. Common method biases in behavioral research: A critical review of the literature and recommended remedies. J. Appl. Psychol. 2003, 88(5), p. 879.

Round 2

Reviewer 2 Report

Comments and Suggestions for Authors

Thank you for careful revision of the manuscript.